

# Search for morphological indicators that predict implantation by principal component analysis using images of blastocyst

Daisuke Mashiko[1], Mikiko Tokoro[1,2], Masae Kojima[2], Noritaka Fukunaga[2], Yoshimasa Asada[2] and Kazuo Yamagata[1]

[1] Graduate School of Biology-Oriented Science and Technology, Kindai University, Kinokawa, Wakayama, Japan
[2] Asada Institute for Reproductive Medicine, Asada Ladies Clinic, Nagoya, Aichi, Japan

## ABSTRACT

**Background:** Although the current evaluation of human blastocysts is based on the Gardner criteria, there may be other notable parameters. The purpose of our study was to clarify whether the morphology of blastocysts has notable indicators other than the Gardner criteria.

**Methods:** To find such indicators, we compared blastocysts that showed elevated human chorionic gonadotropin (hCG) levels after transplantation (hCG-positive group; $n = 129$) and those that did not (hCG-negative group; $n = 105$) using principal component analysis of pixel brightness of the images.

**Results:** The comparison revealed that the hCG-positive group had grainy morphology and the hCG-negative group had non-grainy morphology. Classification of the blastocysts by this indicator did not make a difference in Gardner score. Interestingly, all embryos with ≥20% fragmentation were non-grainy. The visual classification based on this analysis was significantly more accurate than the prediction of implantation using the Gardner score ≥3BB. As graininess can be used in combination with the Gardner score, this indicator will enhance current reproductive technologies.

## INTRODUCTION

Human blastocyst selection for transfer is performed by observing its morphology, considering that blastocyst morphology can be examined non-invasively and that it is a parameter that reflects cell viability and developmental capacity. Currently, the evaluation of blastocysts is based on the developmental stage of the blastocyst development, the cell number in the inner cell mass (ICM), and the trophectoderm (TE) (*Gardner & Schoolcraft, 1999*; *Gardner et al., 2000*). This approach is highly effective; as a result, there have also been attempts to automatically score blastocysts using deep learning methods (*Kragh et al., 2019*; *Khosravi et al., 2019*). However, clinicians and patients have ignored the possibility of a useful selection criterion for embryos. Considering that this criterion focuses on several

Corresponding authors
Daisuke Mashiko,
mashikodaisuke@gmail.com
Kazuo Yamagata,
yamagata@waka.kindai.ac.jp

morphological parameters (the developmental stage and cell number) and provides scores, useful information may be obtained from images of the blastocysts. If morphological evaluation can be performed more accurately, the outcome of single-embryo transfer will be improved. By improving the selection of embryos and success rates of single-embryo transfer, instances of multiple gestations and the associated perinatal complications can be reduced. Several attempts have been made to improve this score by focusing on parameters not mentioned in the Gardner score. For example, adding score "D" implies the presence of degenerative tissue (*Veeck & Zaninovic, 2003*). In addition, ICM is classified as A, B, or C based in Gardner criteria, but a study attempted to add scores "D" and "E" (*Stephenson, Braude & Mason, 2007*). However, in their attempt, the parameters were not searched by comprehensive image analysis; thus, it is still not clear whether the morphology can be sufficiently evaluated.

Principal component analysis (PCA) is an effective method to comprehensively express image features (*Mashiko, Ikawa & Fujimoto, 2017*). PCA is a method used to reduce the dimensionality of datasets, increasing interpretability but at the same time minimizing information loss (*Jolliffe & Cadima, 2016*). The use of PCA in face recognition software (*Kong et al., 2005*) is a supportive example for the use of PCA for expressing image features of blastocysts. If the parameters analyzed using PCA are morphometric (*e.g.*, diameter and aspect ratio), there is a risk of overlooking the notable parameter, but this risk can be avoided by performing PCA on the pixel brightness value and subsequent interpretation of the basis. It is possible to extract notable indicators by classifying the data according to the result of the transfer of blastocysts, searching for principal components with different distributions, and interpreting the basis.

Maternal factors such as age, body mass index (BMI), and anti-Müllerian hormone (AMH) are strongly associated in the full-term development of an embryo. Maternal factors may mask notable parameters when focusing on full-term development. In this study, we searched for useful information from images of the blastocysts to predict the results of pregnancy tests, which can be determined based on the measured value of the human chorionic gonadotropin (hCG) level, an indicator of implantation to eliminate maternal influence as much as possible. The association among gestational age, hCG, and fetal growth can result in less reliable ultrasound-derived pregnancy dating, in particular in women with high or low levels of hCG (*Korevaar et al., 2015*). This comparison could help identify morphological indicators to select blastocysts that tend to be implanted.
The purpose of our study was to clarify whether the morphology of the blastocyst has notable indicators other than the Gardner criteria. Moreover, we aimed to search for morphological indicators by performing PCA of blastocyst stage images and comparing the results of the pregnancy test and principal component to perform a comprehensive evaluation of the images.

## METHODS

### Ethics statement

This study was performed in accordance with the tenets of the Declaration of Helsinki. Identifying information has been removed from all sections of the manuscript, including

the Supplemental Information. The analysis of human embryo movies was carried out with the approval of the Ethics Committee of Asada Ladies Clinic (Approval number: 2021-04), Kindai University (approval number: R2-1-001), and the Japanese Society of Obstetrics and Gynecology (date of approval: 06/30/2021). Informed written consent was obtained from all patients included in the study.

## Patient selection and cycle characteristics

The subjects were women aged 20–49 years who underwent oocyte retrieval after ovarian stimulation from March 2019 to September 2020. Patients who had undergone embryo transfer were selected. Patient and cycle characteristics were the same as those described previously (Kitasaka et al., 2021; Asada et al., 2019).

## IVF and ET protocols and medications

*In vitro* fertilization (IVF) and embryo transfer (ET) protocols and medications were as described previously in Asada et al. (2019). Specifically, all frozen-thawed embryo transfers were performed in artificial hormone replacement cycles, according to the endometrial preparation protocol using transdermal estradiol (Estrana; Hisamitsu Pharmaceutical, Saga, Japan) in combination with chlormadinone acetate (Lutoral; Fuji Pharma, Tokyo, Japan). Estradiol treatment was started on day 2 or 3 of the artificial hormone replacement cycle and endometrial thickness was assessed on days 9 to 11 of the cycle. If the endometrium was ≥7 mm, the frozen-thawed embryo transfer was scheduled. Administration of chlormadinone acetate 6 mg/d was started on day 15 of the cycle. The transfer of blastocyst-stage embryos was performed on day 5, considering the day on which the administration of chlormadinone acetate was started as day 0" (Asada et al., 2019).

## Pregnancy tests

When the urinary hCG level at 14 days after embryo transfer was ≥25 IU/mL, the pregnancy test was considered to be positive.

## Principal component analysis

Images of the blastocysts were obtained using the zygote observation system (CCM-iBIS NEXT; ASTEC Co., Ltd., Fukuoka, Japan) at the Asada Ladies Clinic, and the frame just before freezing or just before transplantation was used for the PCA.

The equatorial planes were focal. All embryos included in the study were ≥ expansion B3, Gardner grade. The images of the zona pellucida were cropped as a circle using ImageJ (https://imagej.nih.gov/ij/), and the brightness value outside the circle was set to 0. To reduce the calculation time using PCA, the cropped images were resized to 64 × 64 pixels to normalize the size (Supplemental Information). The PCA was run at Google Colaboratory (https://colab.research.google.com/). The custom code for the PCA is at GitHub (https://github.com/mashikodaisuke/PCA_code).

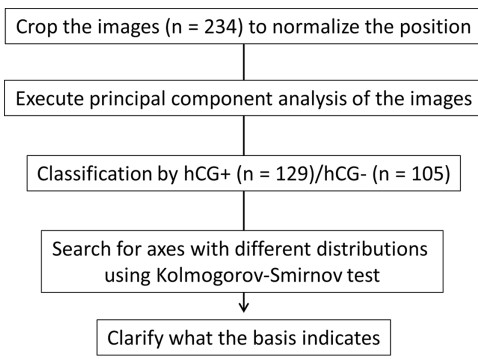

**Figure 1** **Schematic of the experimental strategy.** We cropped the image for normalization; subsequently, we performed principal component analysis (PCA). We searched for axes that separate the human chorionic gonadotropin (hCG)-positive group from the negative group and clarified what the basis indicates.

## Gardner scoring

Blastocysts were scored based on the Gardner criteria by an embryologist at the Asada Ladies Clinic. Blastocysts were evaluated on day 5 using the zygote observation system (CCM-iBIS NEXT; ASTEC Co., Ltd., Fukuoka, Japan), and embryos showing expansion <3 were re-evaluated on days 6 and 7.

## Statistical analysis

Kolmogorov–Smirnov test, McNemer's Chi-squared test, two-sided Welch's $t$-test, and two-sided Student's $t$-test were performed using R (https://www.r-project.org/). Receiver operating characteristic analysis was performed using the rocr package (https://cran.r-project.org/web/packages/ROCR/index.html).

# RESULTS

## PCA using blastocyst images

The images of embryos in the blastocyst stage were cropped to normalize the position of observation, and PCA was performed on the brightness of the pixels of the images. As the embryo was circular and the cropped image was a square, the four corners were trimmed. The positive group ($n = 129$) and negative group ($n = 105$) of pregnancy tests based on the hCG levels were classified, and we searched for axes with different distributions (Fig. 1). As a result, the distribution in principal component (PC)3 (contribution ratio = 4.14%) and PC23 (contribution ratio = 0.64%) were significantly different (Kolmogoro–Smirnov test, PC3: $P = 0.020$, PC23: $P = 0.030$) (Fig. 2, Table S1).

## Blastocyst graininess was the highlighted morphological parameter

Subsequently, we examined the PCs to clarify what morphological parameters they indicated. PC3 emphasized the outer edge of the image and PC23 emphasized each cell (Fig. 3A). Receiver operating characteristic (ROC) analysis was performed on PC3 and

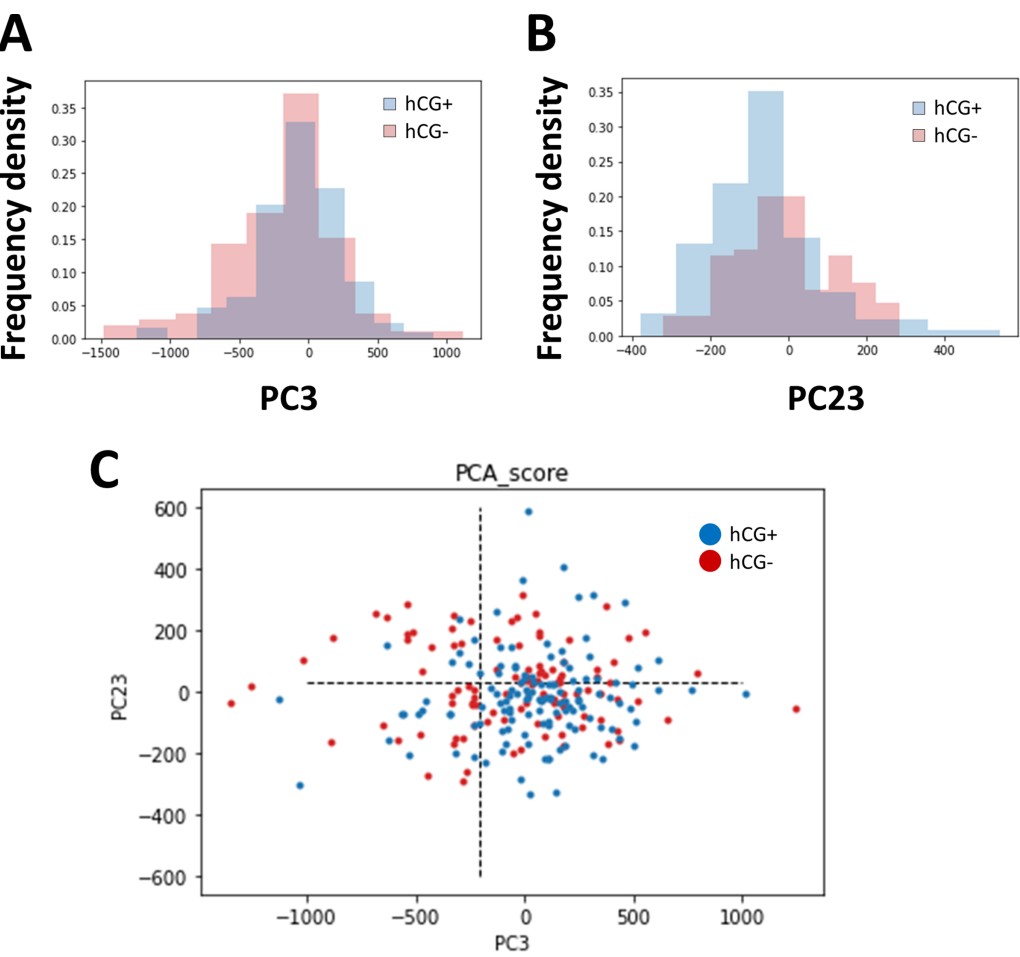

**Figure 2** **Components showing significant differences between the human chorionic gonadotropin (hCG)-positive group and the hCG-negative group and their distribution.** (A) Distribution of the hCG-positive group (cyan) and hCG-negative group (pink) in principal component (PC)3 (PC3). (B) Distribution of the hCG-positive group (cyan) and hCG-negative group (pink) in PC23. (C) Two-dimensional distribution of the hCG-positive group (blue) and hCG-negative group (red) in PC3 and PC23. The dotted lines show the threshold value determined using the receiver operating characteristic (ROC) analysis. We analyzed 196 patients and 234 embryos.

PC23, and the threshold values for dividing each group were calculated (Table S1). The area under the curve (AUC) was 0.58, 0.59, and 0.63 for "PC3," "PC23," and "combination of PC3 and PC23," respectively. By comparing the groups exceeding thresholds (PC3 or PC23) and non-exceeding groups, the blastocysts were classified into grainy and non-grainy (Fig. 3B). In addition, the Gardner scores were compared between the groups that exceeded and did not exceed the thresholds. Considering that the Gardner score consists of three order scales (expansion, ICM, and TE) (*e.g.*, 3BC), we compared each parameter between the groups that exceeded and did not exceed the thresholds. There was no significant difference in the scores of expansion, ICM, and TE ($P = 0.11, 0.22, 0.10$), respectively.

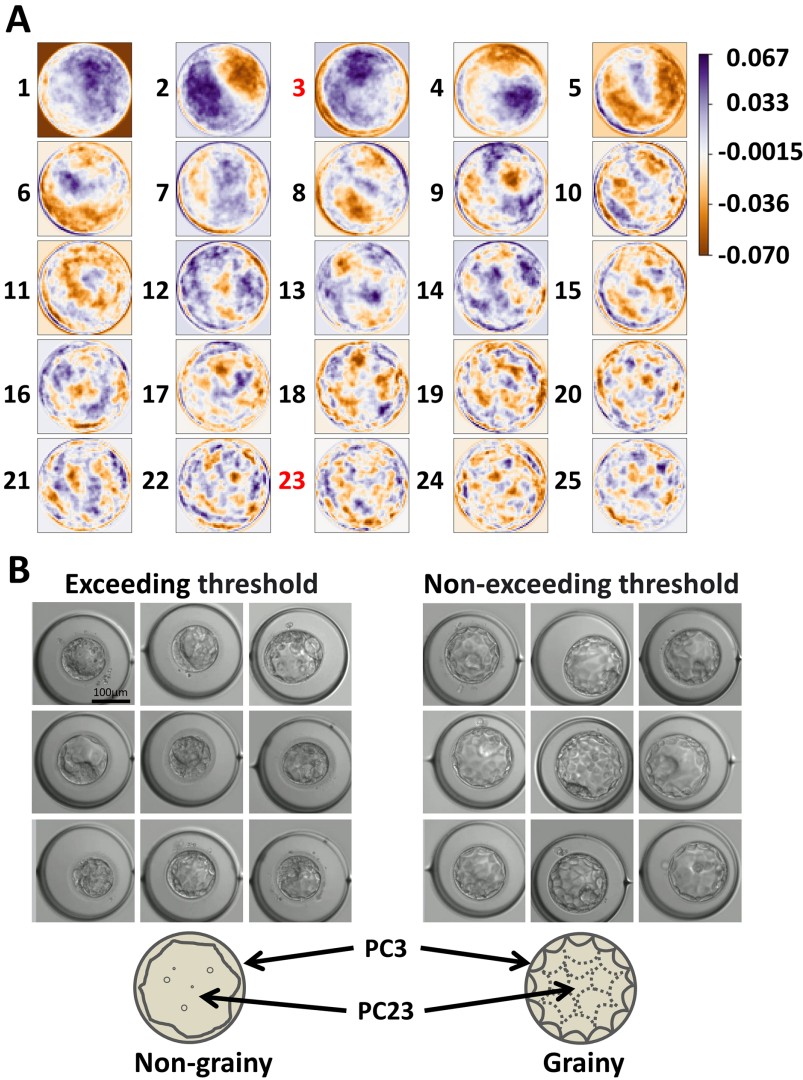

**Figure 3 Parameters emphasized by each basis and the morphology represented by principal component (PC)3 and PC23.** (A) Parameters emphasized by each basis. (B) Typical blastocysts exceeding the threshold and those not exceeding the threshold. The simplified illustrations are below the images. The arrows indicate the parts emphasized by PC3 and PC23. We analyzed 196 patients and 234 embryos.

## Feature of non-grainy blastocysts

Movies were retrospectively analyzed to clarify the development of non-grainy embryos. Non-grainy blastocysts showed fragmentation during mitosis, and cellular debris was observed in the morula stage (Fig. S1 and Movie S1). When the volume of the blastocyst increased, the cellular debris was squashed, making it difficult to distinguish each cell, resulting in a non-grainy blastocyst (Fig. S1). We demonstrated the correlations between graininess and embryo fragmentation. All embryos with ≥20% fragmentation appeared non-grainy (110/110). On the contrary, all embryos with <20% fragmentation did not appear non-grainy (0/124). There was a significant correlation between graininess and embryo fragmentation (prop-test, $P = 2.2 \times 10^{-16}$).

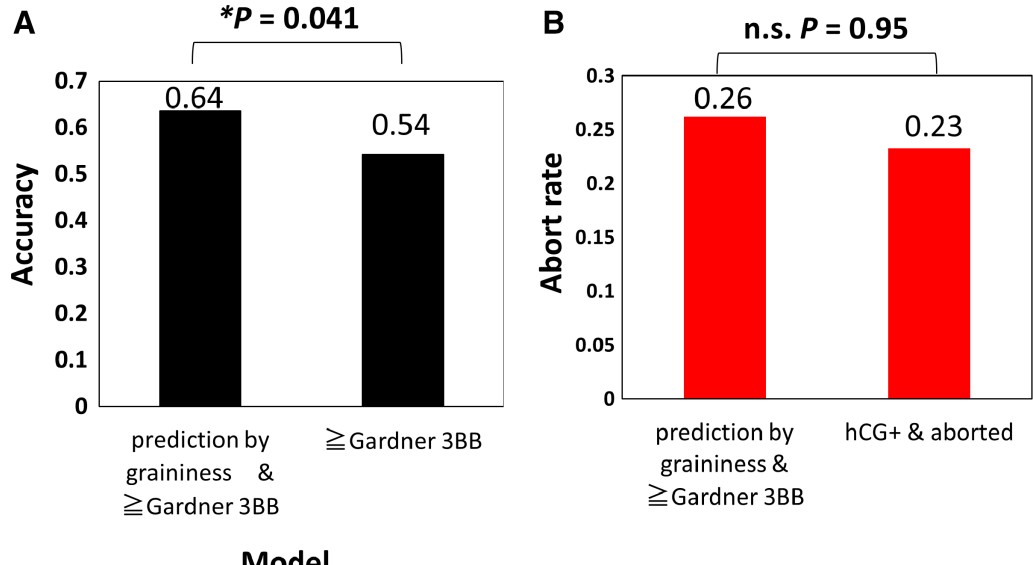

**Figure 4 Prediction of implantation by graininess and Gardner score was higher than that of the Gardner score alone.** (A) Prediction accuracy of implantation by graininess and Gardner score, and prediction accuracy of implantation by Gardner score. (B) Abortion (gestational sac negative) rate after prediction. We analyzed 234 embryos.

## Visual classification of blastocyst by graininess improves implantation (hCG positive) prediction accuracy

Finally, because it is laborious to calculate the PC score from the basis each time, the effectiveness of predicting the result of the pregnancy test by visually dividing it into grainy and non-grainy blastocysts was verified (Fig. 4). The results showed an accuracy of 0.64, precision of 0.68, recall of 0.65, specificity of 0.62, F1-measure (harmonic mean of precision and recall) of 0.66, F0.5-measure (harmonic mean that focuses on precision) of 0.67, and F2-measure (harmonic mean that focuses on recall) of 0.66. In the evaluation of the embryos using the Gardner score, if the score is below 3BB, it is defined as poor quality (*Teranishi et al., 2009*). The accuracy of predicting implantation (hCG positive) with Gardner score ≥ 3BB was 0.54 (precision = 0.55, recall = 0.96, specificity = 0.029, F1-measure = 0.70, F0.5-measure [focus on precision] = 0.60, F2-measure [focus on recall] = 0.84), and the accuracy predicted by graininess was significantly higher than this prediction (Fig. 4A) (McNemer's Chi-squared test, $P = 0.041$). Implantation can be effectively predicted by graininess, but if the subsequent abortion (gestational sac negative) rate is high, it will not be an indicator of embryo selection. Thus, we compared the abortion rate (26%: 22/84) in the prediction group with the actual abortion rate (23%, 32/129), and there were no significant differences (Fig. 4B) (prop-test, $P = 0.95$). In addition, we compared the age of patients in the grainy (33.3 ± 3.1) and non-grainy (34.0 ± 3.3) groups, but there was no significant difference (two-sided Student's *t*-test, $P = 0.21$).

We investigated the correlation between the non-grainy appearance and BMI, AMH, and timing of blastocyst evaluation. BMI ($r = -0.11$), AMH ($r = 0.05$), and timing for blastocyst

evaluation ($r = 0.02$) were not significantly correlated with the non-grainy appearance (test of no correlation; $P = 0.09, 0.44, 0.80$, respectively).

## DISCUSSION

We found the axes that classified the difference between the hCG-positive and negative groups (Figs. 1 and 2). We also revealed that the graininess of the blastocyst was related to the results of the pregnancy test (Fig. 3). In addition, when comparing the visual evaluation based on this analysis with the evaluation based solely on the Gardner score, the accuracy of prediction by graininess was significantly high. We found that there was no subsequent increase in the abortion rate. This indicates that graininess can be an index that can be used for the morphological evaluation of blastocysts.

In recent years, attempts to select good blastocysts have been performed by morphokinetic analysis using time-lapse imaging (*Meseguer et al., 2011*; *Campbell et al., 2013*; *Basile et al., 2014*; *Basile et al., 2015*) and preimplantation genetic testing for aneuploidy (PGT-A) using next generation sequencing (*Neal et al., 2018*; *Patrizio et al., 2019*; *Sato et al., 2019*; *Munné et al., 2019*). On the contrary, research on morphology itself, except for the Gardner criteria, is less active. With the publication of the Gardner criteria, the verification has been actively carried out (*Matsuura et al., 2010*; *Van den Abbeel et al., 2013*; *Hill et al., 2013*; *Subira et al., 2016*), and comprehensive research to search for other parameters has not been conducted. The graininess of the blastocysts, found in the present study, was associated with fragmentation during the early stage (Fig. S1). Fragmentation of embryos in the cleavage stage has been suggested to affect implantation (*Veeck, 1999*; *Alikani et al., 1999*; *Giorgetti et al., 1995*; *Taşdemir et al., 1995*; *Staessen et al., 1992*). Graininess reflects fragmentation in the early cleavage stage and the cellular debris in the morula stage. Evaluation of embryo by graininess is useful because evaluation that reflects fragmentation can be performed without time-lapse imaging.

We compared the parameters of the Gardner criteria between the groups that exceeded the threshold of PC3 and PC23 and those that did not, but no significant difference was found. Furthermore, the definition of the Gardner criteria that was previously described in *Gardner et al. (2000)* is as follows, with no description of graininess: "For blastocysts graded as 3–6 (*i.e.*, full blastocysts onward), the development of the inner cell mass was assessed as follows: A, tightly packed, many cells; B, loosely grouped, several cells; or C, very few cells. The trophectoderm was assessed as follows: A, many cells forming a cohesive epithelium; B, few cells forming a loose epithelium; or C, very few large cells" (*Gardner et al., 2000*). Thus, the graininess of blastocysts can be a notable indicator other than the Gardner criteria.

The PCA and group-to-group comparisons for pixel brightness did not extract the parameters defined by the Gardner criteria. As the Gardner criteria consist of complex parameters such as cell number and cohesive/loose, the parameters that integrate these into the basis were not presented. The PCA did not reveal parameters related to the Gardner score because it is difficult to express the cell number in brightness information. In addition, it was difficult to normalize the location of the ICM; therefore, the location and size of the ICM could not be evaluated using the PCA.

Graininess, which was found as a morphological indicator of blastocysts that can predict implantation in this study, can be useful in selecting embryos for transplantation from multiple blastocysts (Fig. 4). Time-lapse imaging (TLI) is performed at an additional cost, and some researchers do not enthusiastically support the use of TLI in standard care (*Bhide et al., 2020*). Our study showed that there is a possibility that evaluations that reflect the fragmentation of blastocysts in the early cleavage stage can be performed even in clinics that do not perform long-term time-lapse imaging. This morphological predictor has higher accuracy than the Gardner score ≥3BB, indicating that wasteful transplantation can be avoided. However, there will be some potential loss during transfer because it has a lower recall than prediction using the Gardner score ≥3BB. It may be effective to apply this index when the number of blastocysts is high.

## ACKNOWLEDGEMENTS

We would like to thank Editage for English language editing.

### Funding
This work was supported by the JSPS KAKENHI, with grant numbers JP25712035, JP25116005, JP18H05528, and JP18H02357 to Kazuo Yamagata. Daisuke Mashiko is a JSPS Research Fellow (202100155). The funders had no role in study design, data collection and analysis, decision to publish, or preparation of the manuscript.

### Grant Disclosures
The following grant information was disclosed by the authors:
JSPS KAKENHI: JP25712035, JP25116005, JP18H05528 and JP18H02357.
JSPS Research Fellow: 202100155.

### Competing Interests
The authors declare that they have no competing interests.

### Author Contributions
- Daisuke Mashiko conceived and designed the experiments, performed the experiments, analyzed the data, prepared figures and/or tables, authored or reviewed drafts of the paper, and approved the final draft.
- Mikiko Tokoro performed the experiments, analyzed the data, authored or reviewed drafts of the paper, and approved the final draft.
- Masae Kojima performed the experiments, analyzed the data, authored or reviewed drafts of the paper, and approved the final draft.
- Noritaka Fukunaga conceived and designed the experiments, authored or reviewed drafts of the paper, and approved the final draft.
- Yoshimasa Asada conceived and designed the experiments, authored or reviewed drafts of the paper, and approved the final draft.

- Kazuo Yamagata conceived and designed the experiments, authored or reviewed drafts of the paper, and approved the final draft.

## Human Ethics

The following information was supplied relating to ethical approvals (*i.e.*, approving body and any reference numbers):

Analysis of human embryo movies was carried out with the approval of the Ethics Committee of Asada Ladies Clinic (Approval number: 2021-04), Kindai University (approval number: R2-1-001), and the Japanese Society of Obstetrics and Gynecology (date of approval: 06/30/2021).

## Data Availability

The raw measurements are available in the Supplemental Files.

## Supplemental Information

Supplemental information for this article can be found online at http://dx.doi.org/10.7717/peerj.13441#supplemental-information.

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
