# Peer review of "Search for morphological indicators that predict implantation by principal component analysis using images of blastocyst"

_PeerJ, doi:10.7717/peerj.13441_

## Round 0.1 · original submission · Minor Revisions

Both Reviewers have found the manuscript interesting with potential significance. However, some minor issues were also pointed out during the review process. Please refer to the Reviewers' comments for details.

Reviewer 1 ·

Basic reporting

no comment

Experimental design

no comment

Validity of the findings

no comment

Additional comments

The manuscript by Mashiko et al evaluated the utility of classifying blastocyst morphology into grainy and non-grainy for the purpose of successfully predicting implantation or pregnancy more accurately, as indicated by positive pregnancy tests. The results indicated that successful pregnancies had blastocysts with grainy morphology and unsuccessful pregnancies had non-grainy morphology. Additionally, there were no significant difference in Gardner scores between grainy and non-grainy blastocysts, indicating that this morphology criteria can be used independently and/or as a supplement to the Gardner score.
This manuscript is applying for publication in PeerJ’s “The Journal of Life and Environmental Sciences”. Since this journal is not specific to reproductive health and technology, there are some terms that should be defined, however briefly, to be meaningfully to an audience that may not be familiar with reproductive health research.
The introduction highlighted a deficiency of study into classifications outside the Gardner score, placing a spotlight on an important area of research in reproductive health and suggests that complacency with the Gardner score may be blinding individuals, such as clinicians and patients, to the possibility of a useful, if not better, selection criterion for embryos. This could have been stated explicitly to be more informative to individuals who have a vested interest in this research.
After the title, line 38 introduces the reader to principal component analysis (PCA). There was no description of what PCA is other than line 67 mentioning that it is an effective method for expressing image features. It would be informative if the author summarized PCA. The use of PCA in face recognition software would have served as a supportive example for the use of PCA for expressing image features of grainy versus non-grainy blastocysts. Also suggested is to represent the findings more definitively within the abstract, there are some major significant findings (e.g. lines 170-174; 185-192) in the results section not stated in the abstract.
Line 58: what other “notable parameters”?
Line 59: it was stated the outcome of single-embryo transfer will be improved by more accurate morphological evaluation of blastocysts. Perhaps a brief statement on the practical benefits of improving embryo selection and success rates of single-embryo transfer should be included, such as reducing instances of multiple gestation and associated perinatal complications.
Line 75: maternal factors were brought up, but besides subject age, mentioned on line 99, there were no other maternal factors listed. Clarification of what is meant by maternal influences should be included, such as genetic influence, socioeconomic status, employment status, and available social support.
Line 77: what “morphological parameters”? a quick literature search found others.
Line 104: in brief, please provide some information regarding parameters on embryo transfer since it is known that differences can exists in outcomes between techs/physicians
Lines 117-118: all embryos included in the study were ≥3B Gardner grade. A brief rationale for this criterion should be included, such as if this Gardner grade was the industry standard for embryo selection or if it was recommended by an embryologist.
Line 120 states that 64 x 64 pixels was the chosen resolution to normalize image size. Embryo-scopes are capable of higher resolution images, an explanation for the chosen 64 x 64 resolution should have been included.
Lines139-140: it was stated the blastocyst images were cropped to normalize the position of observation. It should be clarified what was cropped out and why, such as if the images were cropped to allow the blastocyst to take up as much of the 64x64 pixel space as possible. Or if there was another rationale for how each image was cropped.
Line 156: In addition to the classification of blastocysts into grainy and non-grainy, the authors should have mentioned if there were other notable parameters considered and rejected. Perhaps a notable criterion such as actual diameter of blastocyst.
Line 218: it was stated research to search for other parameters had not been conducted, but it was stated in the introduction at lines 61-62 that attempts have been made that focused on parameters not included in the Gardner score, that is degenerative tissue. Additionally, there has been research into the presence of cellular fragments, a criterion not used in the Gardner score, in the perivitelline space a predictor of expanded blastocyst quality: https://www.frontiersin.org/articles/10.3389/fcell.2020.616801/full
Lines 245-247 stated clinical implications of evaluating grainy and non-grainy in blastocyst. This criterion would be useful in selecting embryos for transplantation from multiple blastocysts and that this evaluation may be used to reflect fragmentation in clinics that do not perform time-lapse imaging. It would be informative to mention that time-lapse imaging (TLI) is an additional cost and some research does not enthusiastically support the use of TLI in standard care as mentioned in this article: https://trialsjournal.biomedcentral.com/articles/10.1186/s13063-020-04537-2.
As previously mentioned, including brief descriptions of the terms noted in the article would reduce the workload of the reader and enhance the reading of the article. The mention of clinical implications in the abstract would greatly improve the impact of the abstract.

Reviewer 2 ·

Basic reporting

In this manuscript titled “Search for morphological indicators that predict implantation by principal component analysis using images of blastocyst”, the authors came up with a new parameter based on the morphology of blastocyst to evaluate the blastocyst capacity. The results showed the blastocysts from the mothers who showed higher human chorionic gonadotropin (hCG) presented the grainy morphology, which seemed more accurate than the prediction of implantation by Gardner score. Furthermore, authors suggested that graininess also can be combined with Gardener score to improve implantation prediction accuracy.
Generally, the manuscript was written well and data are clearly presented. The finding is interesting for current reproductive technologies. I have several concerns regarding the manuscript in its current form which are listed below.
Major comments:
1. Authors divided the blastocysts into grainy and non-grainy according to exceeding thresholds and non-exceeding thresholds. However, based on the morphology in Figure 3B, grainy blastocysts seemed to show late developmental stage of blastocysts with bigger expansion and cell number as already described in Gardner score. Therefore, the graininess criteria seems similar to Gardner score except more calculation based on images. Authors should explain more and display the typical difference between the graininess criteria and Gardner score.
2. In line 173, authors revealed the correlations between graininess and embryo fragmentation, and hastily concluded that squashed cellular debris caused the failure of graininess. Apparent debris could be the feature or indicator for the failure of graininess, not the mechanisms. Authors should show more solid data, for example add some cellular apoptosis assay, to support this mechanism.
3. In Figure 4, authors only present the summary of prediction accuracy and abortion rate after prediction, which appeared graininess was more accurate in prediction. It would be supportive if authors provide more details about maternal conditions, such as individual age, hormone levels, BMI and times of embryo transfer.
Minor comments:
4. In Figure 2 and Figure 3, the number of patients or embryos should be added in legends.
5. In line 75, some references should be cited when discussed maternal factors like hCG. Besides, this paper (10.1007/s10654-015-0039-0) probably will be helpful to explain why chose hCG.
6. In discussion part, the paragraphs format is different from introduction.

Experimental design

no comment

Validity of the findings

no comment

Reviewer 3 ·

Basic reporting

Attached

Experimental design

Attached

Validity of the findings

Attached

Annotated reviews are not available for download in order to protect the identity of reviewers who chose to remain anonymous.

---

## Round 0.2 · accepted · Accept

All questions and concerns have been well addressed.

Reviewer 2 ·

Basic reporting

no comment

Experimental design

no comment

Validity of the findings

no comment